# The Cellular and Protein Arms of Coagulation in Diabetes: Established and Potential Targets for the Reduction of Thrombotic Risk

**DOI:** 10.3390/ijms242015328

**Published:** 2023-10-18

**Authors:** Nawaz Z. Safdar, Noppadol Kietsiriroje, Ramzi A. Ajjan

**Affiliations:** 1Department of Internal Medicine, St James’s University Hospital, Leeds Teaching Hospitals NHS Trust, Leeds LS9 7TF, UK; n.safdar@nhs.net; 2Light Laboratories, Leeds Institute of Cardiovascular and Metabolic Medicine, University of Leeds, 6 Clarendon Way, Leeds LS2 3AA, UK; 3Endocrinology and Metabolism Unit, Faculty of Medicine, Prince of Songkla University, Songkla 90110, Thailand; noppadol.k@psu.ac.th

**Keywords:** diabetes, thrombosis, antiplatelets, anticoagulant

## Abstract

Diabetes is a metabolic condition with a rising global prevalence and is characterised by abnormally high blood glucose levels. Cardiovascular disease (CVD) accounts for the majority of deaths in diabetes and, despite improvements in therapy, mortality and hospitalisations in this cohort remain disproportionally higher compared to individuals with normal glucose metabolism. One mechanism for increased CVD risk is enhanced thrombosis potential, due to altered function of the cellular and acellular arms of coagulation. Different mechanisms have been identified that mediate disordered blood clot formation and breakdown in diabetes, including dysglycaemia, insulin resistance, and metabolic co-morbidities. Collectively, these induce platelet/endothelial dysfunction and impair the fibrinolytic process, thus creating a prothrombotic milieu. Despite these abnormalities, current antithrombotic therapies are largely similar in diabetes compared to those without this condition, which explains the high proportion of patients experiencing treatment failure while also displaying an increased risk of bleeding events. In this narrative review, we aimed to summarise the physiological functioning of haemostasis followed by the pathological effects of diabetes mellitus on platelets and the fibrin network. Moreover, we carefully reviewed the literature to describe the current and future therapeutic targets to lower the thrombosis risk and improve vascular outcomes in diabetes.

## 1. Introduction

In 2019, the global prevalence of diabetes mellitus was estimated to be 463 million, and this is expected to rise by a staggering 50% by 2045 [1]. Diabetes is a metabolic disease characterised by abnormally elevated blood glucose levels, with the two most common types of diabetes occurring secondary to total insulin deficiency in type 1 diabetes (T1D; around 5–10%), or resistance with relative insulin deficiency in type 2 diabetes (T2D; around 90%) [2]. Cardiovascular disease accounts for a large proportion of excess mortality in patients with diabetes [3], and of those with co-morbid cardiovascular disease, atherothrombosis is the leading cause [4]. Consequently, international guidelines have advocated aggressive cardiovascular disease risk management and optimisation [2]. As a result, there has been a global decline in all-cause and cardiovascular disease-related mortality in those with diabetes [5]. Nonetheless, mortality and hospitalisation secondary to atherothrombotic events remain much higher in patients with diabetes when compared to a matched cohort of patients with normal glucose metabolism [6].

One mechanism for the increased cardiovascular risk in diabetes is the enhanced thrombosis potential coupled with hypofibrinolysis [7]. Despite these known abnormalities, antithrombotic therapies in individuals with diabetes remain largely similar to those with normal glucose metabolism [2]. In this narrative review, we aimed to summarise the physiological functioning of haemostasis followed by the pathological effects of diabetes mellitus on platelets and the fibrin network. Moreover, we systematically reviewed the literature to describe the current and future targets, and the pharmacological agents that may be used for lowering the risk of thrombosis in those with diabetes mellitus. Appendix A details the systematic search strategy to identify relevant studies.

## 2. The Mechanistic Pathways of Coagulation

### 2.1. Physiological Primary Haemostasis

To understand the pathological effects of diabetes on coagulation, let us recap the physiological functions of primary and secondary haemostasis, which are important components of coagulation (Figure 1) [8]. As the endothelium lining blood vessels undergoes damage, tissue factor and collagen are exposed to the subendothelial matrix [9], kickstarting primary haemostasis (Figure 1A). Platelets adhere to the site of injury, mediated by the binding of platelet surface receptor glycoprotein Ib-IX-V (GPIb-IX-V) to von Willebrand factor (vWF) [10]. Specifically, high shear stress from blood flow activates vWF, allowing for its A1 domain to interact with the GPIbα subunit of the GPIb-IX-V receptor of platelets [11]. A secondary binding of the receptor glycoprotein VI (GPVI) to type 1 and type 3 collagen that is exposed from the subendothelium [12] aims to further strengthen platelet adherence to the site of injury.

Subsequent platelet activation exposes binding sites on integrins such as αIIbβ3 (glycoprotein IIb/IIIa; GPIIb/IIIa) to multiple ligands including fibrinogen, vWF, fibronectin, and CD40 [13] via inside-out and outside-in signalling pathways [14]. Once activated, platelets secrete adenosine diphosphate and thromboxane A2 [15], via dense granules, which bind to P2Y_1_/P2Y_12_ and thromboxane receptors, respectively. This positive feedback causes platelet aggregation and contributes to the formation of the platelet clot that subsequently helps with secondary haemostasis, described in the next section.

### 2.2. Physiological Secondary Haemostasis

In addition to the cellular arm, the protein arm of coagulation is also activated following vascular injury [8]. Traditionally, coagulation was divided into the intrinsic and extrinsic pathways (initiated by factor XII and tissue factor, respectively); however, the two pathways are interlinked and usually activated in vivo as a single pathway [16]. When blood vessels are damaged, blood is exposed to tissue factor (TF) located in extravascular tissue (Figure 1B) [17]. Along with its co-factor, factor VIIa, the TF–factor VIIa complex activates factor IX and factor X. Factor Xa then activates thrombin from its inactive precursor prothrombin following interaction with its co-factor Va [18]. The generated thrombin, and also factor XII, activate factor XI, and this starts a downstream cascade: factor IXa is produced, which, alongside factor VIIIa, promotes further conversion of factor X to factor Xa [8]. The additional factor Xa contributes to the cleavage of more prothrombin into thrombin, and the newly activated thrombin mediates the generation of insoluble fibrin by cleaving the soluble plasma protein, fibrinogen.

Primary and secondary haemostases interact when thrombin cleaves protease-activated receptors 1 and 4 (PAR-1/4) [19], exposing an N-terminus, which results in platelet remodelling, integrin activation, and granule secretion via cell-signalling pathways. The insoluble fibrin produced by the coagulation cascade then aids in stabilising the clot by linking platelets and providing a skeleton that embeds blood cellular components [20].

### 2.3. Fibrin and Fibrinolysis

Formation of the fibrin clot begins when insoluble fibrin is cross-linked by factor (F) XIIIa (Figure 1C) to form protofibrils by lateral aggregation of fibrin fibres [21]. Conversely, the dissolution of the fibrin clot is essential to promote further wound healing [22]. As such, tissue plasminogen activator (tPA) and plasminogen, two proenzymes found in the blood, bind to the surface of the fibrin clot [7] along with urokinase. tPA/urokinase activates plasminogen to plasmin, which then cleaves fibrin, thus limiting clot extension.

Fibrinolysis is kept in check by a number of proteins; circulating plasmin and plasminogen activators are regulated by serine protease inhibitors such as plasminogen activator inhibitor 1 (PAI-1) [23], α-2 antiplasmin (α2AP) [24], and complement component 3 (C3) [25]; the former is found circulating within plasma, whereas the latter two are incorporated into the fibrin clot. α2AP and C3 increase the clot’s resistance to lysis and ensure stability, whereas PAI-1 is a specific inhibitor of tPA and urokinase. Further, thrombin aids in the activation of thrombin activatable fibrinolysis inhibitor (TAFI) [26], an antifibrinolytic protein that is cross-linked by FXIIIa into the clot [27]. Its activation protects the fibrin clot by reducing plasminogen and tPA binding [28], thereby limiting plasmin production via a feedback mechanism.

## 3. Platelet Abnormalities in Diabetes Mellitus

### 3.1. Hyperglycaemia

As the abnormal metabolic state of diabetes mellitus continues to progress, platelets undergo pathological changes, which can modulate primary haemostasis (Figure 2). Hyperglycaemia in diabetes induces a state of basal activation and increases their reactivity to stimulants [29,30]. For one, non-enzymatic glycation of platelet membrane proteins results in enhanced expression of adhesion molecules [30]. Platelets from patients with T2D have a greater expression of activation markers (CD31, CD49b, CD62P, and CD63) [31], surface receptors such as glycoprotein Ib and GPIIb/IIIa [32], and surface adhesion molecules including CD40 ligand [33]. CD62P, or P-selectin, plays both an important role in increased platelet activation and in greater platelet adhesion [34]. This impairs the fluidity of the cell membrane and contributes to enhanced reactivity. Further, increased osmolarity during hyperglycaemic states has been found to increase expression of GPVI, GPIIb/IIIa, and P-selectin [35], resulting in enhanced leukocyte linkage to adherent platelets via P-selectin interactions with P-selectin glycoprotein ligand 1 (PSGL-1) [36]. Moreover, protein kinase C, an essential mediator of platelet activation, is also upregulated in response to acute hyperglycaemia [37,38], causing increased activation of platelets. Finally, glycation of low-density lipoproteins (LDL) induces structural changes in LDL and alters their interaction with platelets, increasing intracellular calcium concentration within platelets [39].

### 3.2. Insulin Resistance

Insulin interacts with platelets via the insulin receptor on their cell surface [40]. In physiological states, insulin suppresses platelet agonists such as collagen [41] and reduces platelet aggregation [29]. Additionally, insulin causes the release of plasminogen activator [42] alongside an increase in expression of prostacyclin receptors [43] on the surface of platelets. High levels of insulin-like growth factor-1 (IGF-1) receptors are also expressed on the surface of platelets [44]. Due to a relatively lower expression of insulin receptors when compared to IGF-1 receptors, insulin receptor subunits heterodimerise with IGF-1 to form insulin/IGF-1 hybrid receptors [45], which bind IGF-1 more readily than insulin. Consequently, IGF-1 causes downstream phosphorylation of insulin receptor substrate-1 and -2 (IRS-1/2), which blunts platelet reactivity [46].

In individuals with T2D, platelets begin to exhibit insulin resistance. Decreased activation of insulin/IGF-1 hybrid receptors result in fewer phosphorylated IRS-1 and IRS-2 molecules. As a consequence, intracellular calcium concentration is increased [47], leading to enhanced platelet aggregation and, thus, reactivity. Furthermore, insulin resistance also decreases platelet sensitivity to nitric oxide [48] and prostacyclin [49], molecules which are antagonistic to platelet reactivity via their release from the endothelium; therefore, platelet reactivity is further increased.

### 3.3. Influence of Metabolic Co-Morbidities

T2D is often co-morbid with conditions such as obesity and dyslipidaemia, and together, they contribute to increased systemic inflammation. In those with T2D, studies have found increased concentrations of inflammatory markers and platelet activation molecules [50,51]. When inflammatory cells such as leukocytes interact with platelets via PSGL-1-P-selectin linkage, they enhance platelet adhesion, aggregation, and secretion [52] via leukocyte-released substances such as O2^−^, platelet-activation factor, elastase, and cathepsin G. All these factors combined cause platelet hyperreactivity.

Obesity, its influence on insulin resistance notwithstanding, also affects platelet reactivity. Obese patients have been found to have a higher mean platelet volume [53], which in turn is a predictor of increased cardiovascular risk [54]. They also have higher levels of leptin, which has previously been shown to increase platelet aggregation [55]. Finally, the greater oxidative stress and increase in intracellular calcium concentration also boost platelet reactivity in those with high BMIs [56].

In people with dyslipidaemia, platelet activation is enhanced by phospholipids found in LDL molecules [57]. These oxidised phospholipids increase thrombotic risk via a CD36 receptor pathway. Additionally, the stress induced by the increased levels of cholesterol also dysregulates haematopoiesis, which increases platelet counts. Triglyceride levels are typically increased, and alongside molecules that transport triglycerides in the blood called very-low-density lipoproteins, platelet activation is increased [58]. Conversely, lower levels of high-density lipoproteins have an indirect effect on increased platelet reactivity by causing endothelial dysfunction [59].

### 3.4. Platelet and Endothelial Dysfunction

Like the effects of obesity on platelets, in patients with diabetes, there is an increased concentration of intracellular calcium that augments platelet reactivity. Mechanisms that have been hypothesised to play a role include decreased sarcoplasmic endoplasmic reticulum calcium ATPase (SERCA) [60,61], alterations in the functioning of calcium ATPase [62], and the increased action of the sodium/calcium anti-porter leading to an influx of calcium [63]. Oxidative stress leading to enhanced calcium signalling via the formation of superoxide and reduced nitric oxide are also implicated in increasing calcium levels [64].

Aside from the effects of increased oxidative stress on calcium, the production of oxidants such as hydrogen peroxide [65] and superoxide [66] has also previously been found to increase platelet activation. They act on proteins in the presence of hyperglycaemia to produce advanced glycation end products [67], which contribute to atherosclerosis and platelet aggregation via their actions on the receptor for advanced glycation end products [68] and serotonin receptors [69], respectively. Oxidation further disrupts physiological haemostasis via its effects on endothelial function. There is a reduction in the production of nitric oxide and prostacyclin and, coupled with the decreased sensitivity of platelets to these molecules in those with diabetes that we mentioned earlier, there is a synergistic increase in platelet reactivity.

Ultimately, the increased reactivity of platelets results in an increased turnover of platelets, which is confirmed by the finding of a greater number of younger, reticulated platelets in populations with diabetes [70]. Conventional antiplatelet therapy is often inadequate towards these platelets, and this contributes to their blunted actions on this cohort of patients [71].

## 4. Fibrin Abnormalities in Diabetes Mellitus

Fibrin clot structure is notably altered in those with diabetes, and is associated with the formation of more compact fibrin networks that are resistant to fibrinolysis (Figure 3A) [72,73]. Mechanisms for compact fibrin clots include qualitative and quantitative changes in coagulation factors, including the fibrinogen molecule [7]. Fibrinogen levels are increased in T2D and insulin-resistant states, whereas the protein can undergo post-translational modifications, including glycation and oxidation, that directly affect clot structure [74]. Other studies have suggested that altered thrombin generation in diabetes may also affect the fibrin clot, producing more tightly packed fibrin strands, which have an increased resistance to clot lysis [75,76].

While compact clots are more difficult to break down, the fibrinolytic system is also directly affected in diabetes, further contributing to the hypofibrinolytic state that characterises the condition (Figure 3B). Elevated plasma levels of PAI-1 [77], which is a major inhibitor of fibrinolysis, and α2AP [78,79], which is a plasmin inhibitor, have been identified in those with diabetes, whereas increased incorporation of α2AP and C3 into fibrin clots has been documented in patients with T1D [80,81,82], increasing clot resistance to lysis. Finally, high glucose levels can result in plasminogen glycation that affects conversion to plasmin and also compromises the enzymatic efficiency of the protein [83]. Circulating levels of other antifibrinolytic proteins such as TAFI have been found to be significantly greater in those with diabetes when compared to healthy controls, particularly in patients with obesity [84], and higher levels of TAFI correlate with greater thrombotic risk [85,86], including in those with poor glycaemic control [87].

The clinical importance of hypofibrinolysis in diabetes has been emphasised by demonstrating a relationship between adverse clinical outcome and prolonged fibrin clot lysis. In a cohort of over 900 T2D patients with acute coronary syndrome, longer clot lysis was associated with increased cardiovascular mortality after the coronary ischaemic event, despite treatment with dual antiplatelet agents [88]. Therefore, targeting hypofibrinolysis represents one strategy to reduce residual thrombosis risk in individuals with diabetes and established vascular disease.

Interestingly, it is not only hyperglycaemia that results in altered clot structure/lysis, as low glucose levels also have a detrimental effect. Using clamp studies, it was demonstrated that hypoglycaemia prolongs fibrin clot lysis compared with euglycaemic conditions [89]. The abnormality in clot lysis persisted for one week following the hypoglycaemic event, indicating that hypoglycaemia has a long-lasting effect on thrombosis potential, and this has clear clinical implications.

## 5. Current Antithrombotic Therapeutic Options in Diabetes

In this section, we will only discuss studies that included individuals with, or at risk of, atherothrombotic disease and in sinus rhythm, as those with cardiac arrythmias, more specifically atrial fibrillation, require anticoagulant therapies, which is beyond the scope of this review. It should be noted that antithrombotic therapies, whether directed against the cellular or the protein arm of coagulation, increase the risk of bleeding events; therefore, the use of these agents should involve a careful assessment of the balance between the benefit (thrombosis prevention) and the risk (bleeding). This benefit/risk ratio of antithrombotic agents can vary widely in individuals with diabetes and is difficult to accurately assess, given the absence of a reliable test, adding to the complexity of routine clinical care. Therefore, the identification and development of new antithrombotic agents with limited bleeding risk remain key aims of researchers in order to reduce vascular occlusion in diabetes.

Antiplatelet therapies remain the main agents used as antithrombotic therapies to prevent vascular occlusion in those in sinus rhythm. The two main agents used target the thromboxane synthesis pathway (aspirin) or inhibit the G-protein coupled P2Y_12_ receptor (clopidogrel, prasugrel, and ticagrelor). Table 1 summarises current antithrombotic and anticoagulant regimes.

### 5.1. Primary Prevention

Primary prevention includes individuals without a previous history of a vascular event but who may still be at risk, such as individuals with diabetes.

Aspirin is an inhibitor of cyclooxygenase-1 and, thus, reduces thromboxane-A2 production, consequently preventing platelet aggregation. In the past, aspirin was commonly used in patients with diabetes and no clinical cardiovascular disease for primary prevention of thrombotic events, albeit without much evidence. Indeed, several older trials looking specifically at primary prevention failed to find any benefit of aspirin [99,100,101]. Notably, these results were inconclusive due to the small populations analysed and low rate of cardiovascular events, making the studies underpowered [102].

More recently, in a trial involving over 15,000 participants with diabetes without clinical cardiovascular disease [90], aspirin was found to significantly lower the incidence of vascular events. However, the authors concluded that on the balance of the increased risk of bleeding, the benefits were attenuated and therefore limited. This has been reflected in the 2023 ESC guidelines on diabetes and cardiovascular disease, indicating that aspirin use for primary prevention in diabetes “may be considered” (class IIa, level A), and, therefore, starting this therapy will remain at the discretion of the attending physician, after carefully weighing the risks and benefits [2].

Interestingly, data suggest that a once-daily dose of aspirin inadequately inhibits COX-1 and TXA-2 synthesis [103,104]. Using a twice-daily approach, aspirin has been shown to reduce the hyperreactivity seen in platelets from individuals with diabetes more effectively than a once-daily regime [105,106], but it is unclear at present whether this translates into clinical benefit.

### 5.2. Secondary Prevention

This includes individuals with a history of vascular ischaemia. Deciding on the type of antithrombotic strategy is very much dependent on the type of ischaemic event, the timing, and the procedure undertaken to unblock the blood vessel. This is discussed elsewhere in detail [2,107] and only a brief summary is provided here.

For individuals with diabetes and a history of a coronary artery event and who have undergone percutaneous coronary intervention (PCI), dual antiplatelet therapy (DAPT) is recommended, usually with aspirin and ticagrelor or aspirin and prasugrel, for a period of one year, followed by aspirin monotherapy long-term. In higher-risk subjects, DAPT is continued for up to 3 years using aspirin and low-dose ticagrelor [97]. In individuals with diabetes undergoing coronary artery bypass grafting (CABG) following an acute event, DAPT is given only for 12 months, and extended therapy is not recommended due to lack of efficacy data. In the same individuals with chronic coronary artery disease requiring PCI, only 6 months of DAPT is recommended (usually with aspirin and clopidogrel) followed by long-term aspirin monotherapy, whereas in those undergoing CABG, only aspirin is used following the procedure [2].

In patients with diabetes who had an occlusive cerebrovascular event treated with thrombolysis, aspirin is usually used for 3 months followed by long-term clopidogrel monotherapy. If thrombolysis is not used, DAPT is given for a short period (3 weeks) followed by clopidogrel monotherapy [107]. For peripheral artery disease, clopidogrel is the preferred antiplatelet agent for long-term use.

While the above therapies are effective at reducing further thrombotic vascular occlusion, they have two drawbacks: (i) some individuals continue to suffer ongoing thrombotic episodes despite implementation of the above regimes (i.e., treatment failure) and (ii) there is an increase in bleeding risk. Therefore, a more targeted approach is required that normalises the pathological changes of normal haemostasis, thus reducing thrombosis risk without increasing bleeding events.

### 5.3. Current Therapeutic Options in Diabetes: Anticoagulants

Low-molecular-weight or unfractionated heparin, an indirect inhibitor of factor X by its action on antithrombin, is commonly used in acute coronary syndrome, regardless of diabetes status, for its benefits on survival and major adverse cardiovascular events [108]. An alternative drug, fondaparinux, which also acts on factor X, has a better safety profile [109] but does not have any action on thrombin [110]. Newer oral anticoagulants that include factor Xa inhibitors such as rivaroxaban, apixaban, and edoxaban, and thrombin inhibitors such as bivalirudin, represent other options that target the different aspects of secondary haemostasis. For instance, although bivalirudin was found to have similar cardiovascular benefits when compared to heparin and GPIIb/IIIa inhibitors, it had a better bleeding side effect profile [111,112].

The COMPASS trial [98] has demonstrated that the combination of aspirin and low-dose rivaroxaban is associated with better cardiovascular outcomes when compared to aspirin monotherapy and this treatment is particularly effective in the presence of peripheral arterial disease [113], explaining recent guidelines advocating the use of this strategy in higher vascular risk groups, provided bleeding risk is acceptable [2].

## 6. Potential Targets and Future Directions

### 6.1. Antiplatelet Therapies

While antiplatelet therapies have helped to reduce atherothrombosis, we continue to use a “one size fits all” approach, and the current strategy requires some refinement to optimise antiplatelet therapies. For example, evidence suggests a variable sex-linked response to antiplatelet therapies, and this requires further understanding and evaluation [114]. More effective thrombosis and bleeding risk scores are required to further optimise antiplatelet therapies, including the use of combination treatment. Figure 4 is a visual representation of current and future therapeutic targets.

Thromboxane and P2Y_12_

In terms of diabetes-specific areas, and given the increased platelet turnover in this condition and improved response to twice daily aspirin [115], clinical studies are required to understand whether twice daily dosing of aspirin translates into improved clinical outcome (both primary and secondary prevention). Moreover, it remains unclear whether the use of a P2Y_12_ inhibitor monotherapy for long-term prevention is superior to aspirin as some biochemical data suggest this may well be the case [116]. Also, the interaction between glycaemia and the response to antiplatelet therapies requires further research [117], in order to optimise antithrombotic management in diabetes. In short, there is still a significant amount of work to do with existing antiplatelet agents to optimise their use, before moving to new antiplatelet targets outside the thromboxane and P2Y_12_ pathways.

2.Glycoprotein VI (GPVI)

GPVI aids in platelet adhesion to the site of vascular injury, and data suggests that inhibition of GPVI reduces thrombosis with little effect on bleeding time [118]. It is a promising target for antiplatelet therapy due to its isolated expression on platelets and megakaryocytes. A novel GPVI antagonist, revacept, was found to be well-tolerated [119] and showed a small beneficial effect on collagen-induced platelet aggregation [120]. However, only around 25% of the patients had diabetes, and further data are needed in subgroup analyses regarding its benefits.

### 6.2. Anticoagulant Therapies

A consistent abnormality in diabetes is the presence of a hypofibrinolytic environment by mechanisms described above under “fibrin abnormalities in diabetes mellitus”. Therefore, given this abnormality and the association of hypofibrinolysis with adverse vascular outcomes, improving clot lysis represents a credible strategy to reduce vascular events with limited risk of bleeding. Potential targets include:Plasmin inhibitors

Inhibition of the increased concentration and incorporation of α2AP seen in diabetes represents a potential therapeutic modality to reduce thrombosis risk. Indeed, monoclonal antibodies against α2AP have been developed [121] and an ongoing trial is investigating α2AP as a treatment for pulmonary embolism (NCT05408546). An advantage of targeting α2AP is the potential low risk of bleeding, given the targeting of a specific abnormality in diabetes rather than suppression of an entire pathway, which is the case with current antiplatelet and anticoagulant agents.

2.Thrombin activable fibrinolysis inhibitor (TAFI)

The elevated levels of the antifibrinolytic protein TAFI in diabetes may be targeted to reduce thrombotic risk. Despite some positive data, normoalbuminuric and normotensive patients with diabetes were found to have similar levels of TAFI versus controls, a finding that might be explained by the heterogeneous patients included [7]. Nonetheless, conflicting data regarding the utility of TAFI as a target for the development of future therapy require further investigation.

3.Complement component 3 (C3)

C3 is one of the proteins found in fibrin networks with an increased presence in diabetes [7]. Therefore, C3 could represent a diabetes-specific target and, indeed, inhibiting C3’s incorporation into clots has been shown to facilitate clot lysis, particularly in samples from individuals with diabetes [122]. However, the high circulating levels of this protein and its role in innate immune responses may make it problematic as a therapeutic target.

4.Plasminogen activator inhibitor 1 (PAI-1)

PAI-1 plasma levels are classically elevated in T2D and insulin-resistant states [77,123]. A number of strategies have been explored to inhibit PAI-1 function but none have made it to the clinical arena [77], and this remains an area for future research.

5.Protease-activated receptor 1 (PAR-1)

PAR-1 levels have been found to be elevated in those with diabetes [124] and positively correlated to increasing glycated haemoglobin. A therapy aimed at the thrombin receptor, vorapaxar, which selectively binds PAR-1 [125], showed that there was no benefit in terms of a reduction of major adverse cardiovascular events at an increased risk of bleeding [126]. Nonetheless, in a follow-up trial subgroup analysis, where vorapaxar was added onto existing antiplatelet therapy in those with diabetes versus those without [127], there was a significant improvement in major adverse cardiovascular events, albeit at the cost of a higher risk of bleeding. Given the questionable benefit of this agent and the increased risk of bleeding, its clinical use is very limited, and further work is needed to identify alternative therapies targeting PAR-1.

### 6.3. Other Therapies

Platelet-type 12-(S)-lipoxygenase (12-LOX), an oxygenase produced alongside COX-1, potentiates platelet activation via formation of 12(S)-hydroxy-5,8,10,14-eicosatetraenoic acid (12-HETE) and selective 12-LOX inhibition; it has the potential to decrease thrombosis without increasing the risk of bleeding [128]. The role of 12-LOX is linked to the pathogenesis of diabetes mellitus [129], and, therefore, it may represent a future target for the reduction of thrombosis in diabetes.

## 7. Conclusions

Diabetes is associated with an elevated potential for thrombosis due to increased platelet activation and limited response to antiplatelet therapies, as well as the formation of compact fibrin networks and compromised fibrinolysis. Despite these diabetes-specific abnormalities, antithrombotic therapies remain largely similar in diabetes compared to those with normal glucose metabolism. Future research is needed to accurately classify thrombosis and bleeding risk in individuals with diabetes, which must be able to adjust for age, gender, diabetes duration, and presence of co-morbidities. While some mechanistic pathways for the increased risk of thrombosis in diabetes have been elucidated, further research is required to uncover new mechanisms.

More work regarding the optimisation of existing antiplatelet agents, including investigating the effects of multiple daily doses of aspirin and the use of P2Y_12_ inhibitors, rather than aspirin, for primary and long-term secondary prevention, is needed. Despite the disappointment with the PAR-1 inhibitor vorapaxar, future work is still needed to develop new antiplatelet and anticoagulant inhibitors. This includes inhibitors to GPVI, α2AP, TAFI, and PAI-1. “Normalising” the abnormalities encountered in diabetes, using a person-specific approach, will help to reduce thrombotic vascular occlusive events while limiting bleeding risk.

Despite the large amount of work undertaken in this area, we have only scratched the surface, and further research is needed to optimise antithrombotic management in diabetes. This will, in turn, help to improve the adverse vascular outcomes seen in diabetes, which remain the main cause of morbidity and mortality in this high-risk population.

## Figures and Tables

**Figure 1 ijms-24-15328-f001:**
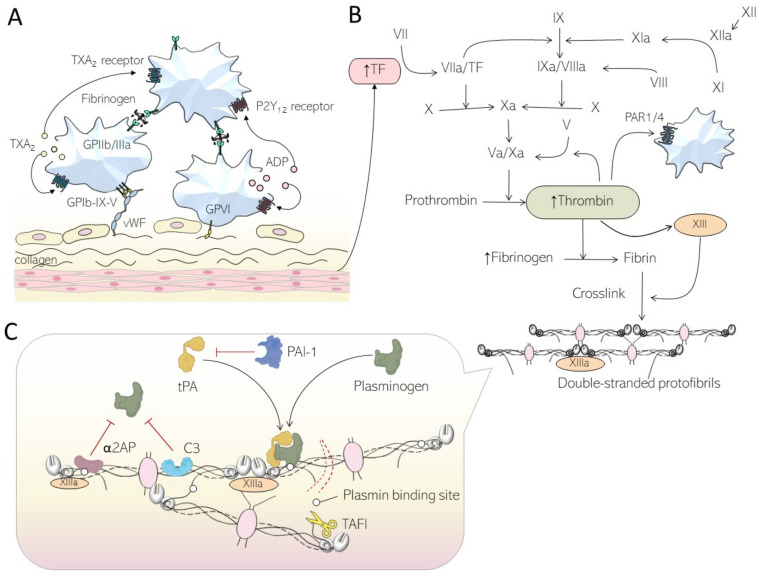
The physiological process of primary and secondary haemostasis. (**A**) Tissue injury exposes molecules like collagen and tissue factor (TF) from the subendothelial matrix, starting primary haemostasis. Platelets begin to adhere to the endothelium via interactions between glycoprotein Ib-IX-V (GPIb-IX-V) and von Willebrand factor (vWF). Glycoprotein VI (GPVI) further strengthens adherence. Platelet aggregation occurs via adenosine diphosphate (ADP)- and thromboxane A_2_ (TXA_2_)-led activation of other platelets by their action on P2Y_12_ and TXA_2_ receptors, respectively, leading to clot formation. Fibrinogen links glycoprotein IIb/IIIa (GPIIb/IIIa) on platelet surfaces to further increase aggregation. (**B**) TF binds to factor VIIa (FVIIa) and starts the downstream cascade of coagulation proteins in secondary haemostasis. The resultant complex of factors Va and Xa cleave prothrombin to form thrombin. This is upregulated, which increases conversion of fibrinogen to fibrin, and the subsequent cross-linking of fibrin into a compact clot. Thrombin also upregulates protease-activated receptor 1 or 4 (PAR1/4), leading to enhanced platelet activation. Antifibrinolytic proteins are cross-linked into the clot, thus increasing resistance to lysis. (**C**) Tissue plasminogen activator (tPA) binds to plasminogen on the fibrin clot, and the resultant plasmin cleaves fibrin to limit clot extension. Plasminogen activator inhibitor 1 (PAI-1) limits the activation of tPA, whereas α-2 antiplasmin (α2AP) and complement component 3 (C3) inhibit plasminogen. Thrombin activatable fibrinolysis inhibitor (TAFI) decreases binding of plasminogen and tPA at the binding site on fibrin, further blunting fibrinolysis. Arrows and blunt arrows represent positive and negative feedback, respectively.

**Figure 2 ijms-24-15328-f002:**
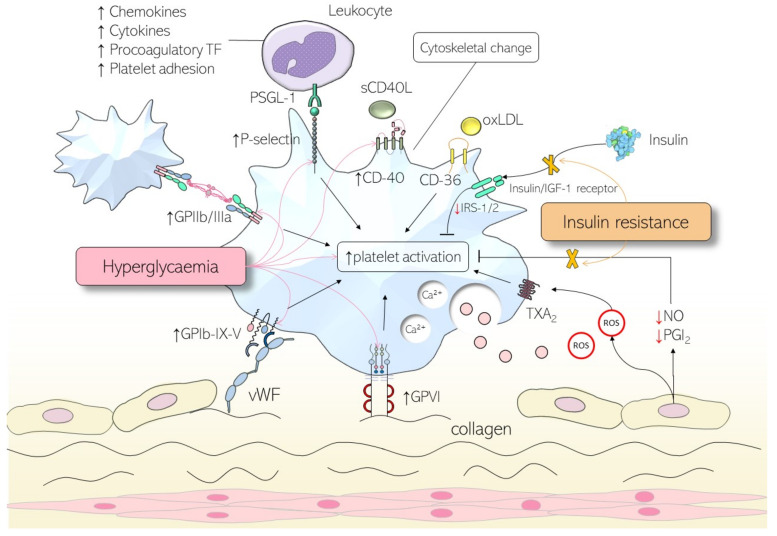
The pathological dysfunction of platelets in diabetes. Hyperglycaemia results in decreased fluidity of cell membrane and increased osmolarity, which increases expression of membrane proteins such as glycoprotein Ib-IX-V (GPIb-IX-V), glycoprotein VI (GPVI), glycoprotein IIb/IIIa (GPIIb/IIIa), P-selectin, and CD-40, and therefore reactivity. The inhibitory effects of insulin on platelet activation, via action on hybrid insulin/insulin-like growth factor 1 (IGF-1) receptors, are also attenuated due to insulin resistance. Increased oxidative stress, lipid dysfunction, upregulation of intracellular calcium, pro-inflammatory cytokines, and reduced production of suppressors of platelet reactivity by the endothelium, including prostacyclin (PG-I_2_) and nitric oxide (NO), are other mechanisms for increased reactivity. Abbreviations: PSGL-1, P-selectin glycoprotein ligand; oxLDL, oxidised low-density lipoprotein; vWF, von Willebrand factor; ROS, reactive oxygen species; TXA_2_, thromboxane A2; IRS-1/2, insulin receptor substrate 1/2; sCD40L, soluble CD40 ligand. *Arrows and blunt arrows represent positive and negative regulation, respectively*.

**Figure 3 ijms-24-15328-f003:**
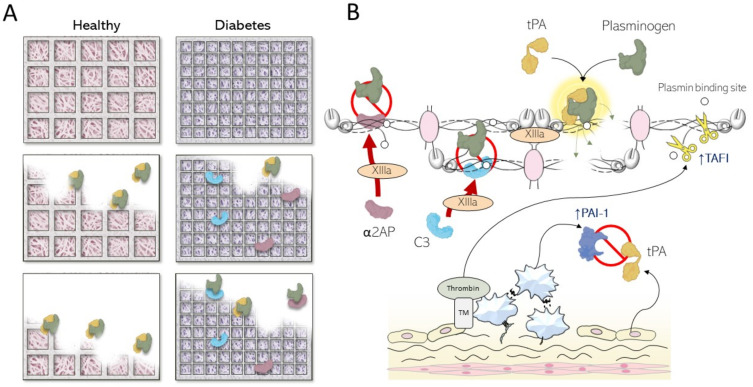
The pathological dysfunction of the fibrinolytic system in diabetes. (**A**) When compared to clots in healthy individuals, people with diabetes have increased antifibrinolytic potential evidenced by the increased resistance to lysis following enhanced incorporation of antifibrinolytic proteins. (**B**) Quantitative and qualitative upregulation of antifibrinolytic and pro-coagulant proteins in diabetes increases resistance of the fibrin clot to fibrinolytic molecules. These include raised levels of plasminogen activator inhibitor 1 (PAI-1) levels, which inhibit plasminogen conversion to plasmin by tissue plasminogen activator (tPA), as well as increased incorporation of antifibrinolytic proteins into the clot such as α2-antiplasmin (α2AP) and complement component 3 (C3). Thrombin activatable fibrinolysis inhibitor (TAFI) levels are also increased in diabetes, further compromising fibrinolysis, while plasminogen is glycated, which reduces conversion to plasmin and alters enzymatic activity of the protein. *Arrows and blunt arrows represent positive and negative regulation, respectively*.

**Figure 4 ijms-24-15328-f004:**
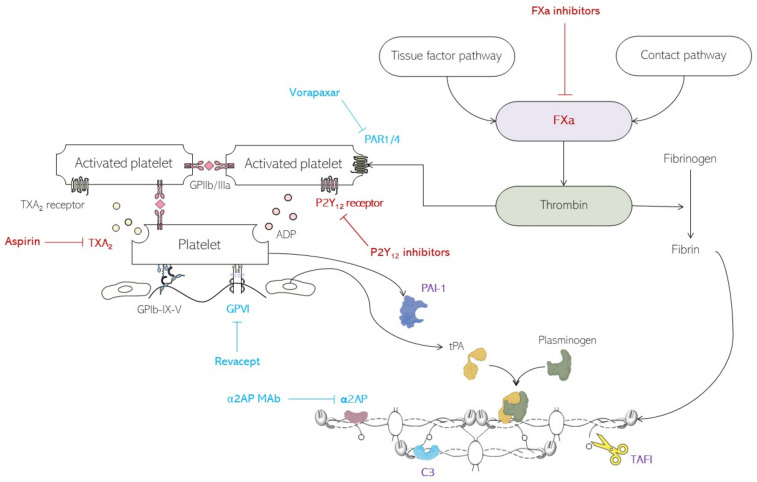
Several current and future therapeutic options exist for the reduction of atherothrombosis risk in diabetes. Aspirin and P2Y_12_ inhibitors are established platelet-specific therapies, whereas factor Xa inhibitors represent alternative or adjunctive therapies aimed at the coagulation cascade to reduce thrombosis risk in diabetes; highlighted in red. Newer agents currently undergoing investigation are highlighted in blue; the protease-activated receptor 1 (PAR-1) antagonist vorapaxar, the glycoprotein VI (GPVI) antagonist revacept, and monoclonal antibodies (MAb) against a plasmin inhibitor α2-antiplasmin (α2AP). Novel and diabetes-specific targets include antifibrinolytic proteins such as complement component 3 (C3), plasmin activator inhibitor 1 (PAI-1), and thrombin activable fibrinolysis inhibitor (TAFI); these are highlighted in purple. *Arrows and blunt arrows represent positive and negative regulation, respectively*.

**Table 1 ijms-24-15328-t001:** Current antiplatelet and anticoagulant therapies in patients with diabetes. OD, once daily; BD, twice daily.

Therapy	Target	Recommended Regime
Antiplatelet agents		
Aspirin	Irreversible inhibitor of COX-1-dependent thromboxane A2 synthesis	Primary prevention with aspirin 75–100 mg OD therapy [90]
		Secondary prevention with aspirin 75–100 mg OD for chronic/acute coronary syndromes [91,92]
Clopidogrel	Irreversible P2Y_12_ inhibitor	Clopidogrel 75 mg monotherapy in aspirin intolerant patients OR dual therapy (clopidogrel 75 mg OD and aspirin 75–100 mg OD) following acute vascular ischaemia
		Prasugrel or ticagrelor superior to clopidogrel in acute coronary syndrome [93,94]
Prasugrel	Irreversible P2Y_12_ inhibitor	Not used as monotherapy
		Dual therapy (aspirin 75–100 mg OD and prasugrel 60 mg) for 12 months in acute coronary syndrome [95]
Ticagrelor	Reversible P2Y_12_ inhibitor	Not used as monotherapy
		Dual therapy (aspirin 75–100 mg OD and ticagrelor 90 mg BD) for 12 months [96]; consider extension to 36 months using low dose (60 mg BD) if no major bleeding risk [97]
**Anticoagulant agents**		
Rivaroxaban	Direct factor Xa inhibitor	Secondary prevention with aspirin 75–100 mg OD and rivaroxaban 2.5 mg BD in chronic coronary syndrome patients with low bleeding risk [98]
		No recommendation in acute coronary syndrome

## Data Availability

Not applicable.

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
