# Peer review of "The Cellular and Protein Arms of Coagulation in Diabetes: Established and Potential Targets for the Reduction of Thrombotic Risk"

_ijms, 2023, doi:10.3390/ijms242015328_

Round 1

Reviewer 1 Report

You have written a comprehensive review of potential mechanisms and therapies associated with thrombosis in patients with diabetes. 

You might however further discuss the bleeding risk of the potential, new therapies . You might also wish to review the potential of 12 lipoxygenase inhibitors  some of which hold the promise of reducin g thrombosis without an increase in bleeding .

Another potential therapeutic approach to reduce thombosis in patients with DM would be to target the intestinal biome. In patients with DM , especailly nthose with long standing hypertension , CKD , HF and the elderly there is an increase in intestinal permeabilityresulting in an increase in LPS, TMAO and a decrease in short chain fatty acids. These systemic abnormalities have been linked to thrombosis . Thus targeting intestinal dysfunction may be another approach to reduce thrombosis in patients with DM.

Author Response

Title: The cellular and protein arms of coagulation in diabetes: established and potential targets for the reduction of thrombotic risk

We would like to take this opportunity to thank the expert reviewers whose time and suggestions have contributed to improving the manuscript and have strengthened the message. We have carefully revised the manuscript according to their comments and an itemised response to each comment and the changes as they appear in the manuscript are provided below.

Reviewer 1

Thank you for your supportive comments and helping us improve not only the message but also how it is presented.

Comment 1: You have written a comprehensive review of potential mechanisms and therapies associated with thrombosis in patients with diabetes. 

Response 1: Thank you.

Comment 2: You might however further discuss the bleeding risk of the potential, new therapies. You might also wish to review the potential of 12 lipoxygenase inhibitors some of which hold the promise of reducing thrombosis without an increase in bleeding.

Response 2: Thank you for the reviewer’s comment. We have added an additional section 6.3 to briefly mention 12-LOX inhibition. Additionally, we have also addressed the bleeding risk under section 5.

Comment 3: Another potential therapeutic approach to reduce thrombosis in patients with DM would be to target the intestinal biome. In patients with DM, especially in those with long standing hypertension, CKD, HF, and the elderly there is an increase in intestinal permeability resulting in an increase in LPS, TMAO and a decrease in short chain fatty acids. These systemic abnormalities have been linked to thrombosis. Thus, targeting intestinal dysfunction may be another approach to reduce thrombosis in patients with DM.

Response 3: This is certainly an interesting suggestion but perhaps beyond the scope of the current review. We feel that this is an important and expanding area of work that would benefit from a separate review article; we would run the risk of not doing the topic justice by briefly mentioning it in the current review. We are open to further suggestions though in case the reviewer feels this topic is key to the current review. 

Reviewer 2 Report

This review addresses a timely topic, namely the prevention of cardiovascular complications of diabetes due to alterations of the hemostatic response. This review has three clearly delimited parts. The first part is a brief account of the physiology of the hemostasis process. It is a good introduction, more or less at text book level, but not more is really needed to set the scene. This part is followed by a summary of the abnormalities in the process caused by diabetes. The last part is about current and prospective therapeutic options. 

The topic is timely, as I mentioned, and in continuous evolution, therefore this review is of value for researchers in the field. The review is well structured and generally well written, and the figures are informative and of good quality. There are nevertheless a number of issues that need to be addressed before I can recommend publication.

1.     The last two paragraphs of the introduction feel disruptive. I suggest moving all this information to the appendix. It only needs a very short reference to the appendix at the end of the introduction.

2.     Section 2, Mechanistic pathways of what? Please complete the title of this section.

3.     Reference to figure 1 should appear much earlier, not nearly at the end of the section. In fact, each individual panel should be referenced in its corresponding subheading (Fig. 1A in section 2.1; Fig. 1B in section 2.2; Fig. 1C in section 2.3). Panel C could be more specifically dedicated to fibrin formation/fibrinolysis and, for completeness, section 2.3 should be expanded a bit. I miss things like urokinase and TAFI (this one is mentioned later, but it should have been introduced here), to mention a few.

4.     In 3.1, CD62P and P-selectin are the same thing but they are presented as activation marker first and as surface adhesion molecule next. Consider re-writing this part, but this is not critical.

5.     In 3.2, 4th line, please complete: …expression of prostacyclin receptors (43) …

6.     In the next paragraph, complete: …receptors result in fewer phosphorylated IRS-1 and…

7.     In the same paragraph the authors mention nitrous oxide, they surely mean nitric oxide. The same happens on pg.5, near the end.

8.     On pg. 5 paragraph 3, the LDL abbreviation was actually introduced earlier. Then you abbreviate VLDL but not HDL. In section 3.4 it may be advisable to include the acronym for SERCA.

9.     Section 4. Please complete the title of this section, Fibrin abnormalities in diabetes mellitus

10.  Figure 2 should be mentioned at the beginning of the relevant section rather than at the very end, so readers can look at it while reading through the information. Same with figure 3. I notice panel B of Figure 3 is not described in the legend. Panels A and B should be referenced separately in the text where they fit best.

11.  Last sentence of pg. 7, please complete: …the thromboxane synthesis pathway… the G-protein coupled receptor P2Y12 (clopidogrel…

12.  Table 1 needs careful consideration. First of all, it includes antithrombotic therapies but also anticoagulants, so either change the title or split into two (antithrombotics, anticoagulants). Clarify in the table heading what OD and BD means. For aspirin, complete: thromboxane A2 synthesis. Several of the therapies are described as No recommendation, so why are they included in a table of current therapies?. Perhaps clarify (as you indicate in the text) that some of them are used in acute coronary syndrome irrespective of diabetic state? I also notice that some (abciximab, eptifibatide, tirofiban, maybe a couple of them more) are not mentioned at all in the text.

13.  On pg. 9, end of 5.1, complete …TXA-2 synthesis.

14.  In 5.2 you use DAPT and CABG acronyms but do not introduce them a couple of sentences before.

15.  On page 10, thromboxane subheading, complete: …use of a P2Y12 inhibitor monotherapy… In the GPVI subheading, you mention collagen 3 “circulating” near the subendothelium; probably not the best word choice, I guess you mean exposed collagen, please re-write.

16.  In the Anticoagulant therapies section of pg 11 (mislabeled 2.1) for each component you include information about how they are altered in diabetes. In most cases this information is mentioned here for the first time. All this should have been included in section 4 and there would be no need to repeat. It may not be necessary to create subheadings for each component; the same applies for section 6.1

17.  In general, when you mention things like described above/below it may be better to indicate exactly which heading or subheading you mean.

18.  Figure 4, unless I missed it, is not referenced in the text. Some of the therapies shown here go with a different color but this is not explained in the legend. I suggest also coloring novel targets like C3, TAFI, PAI-1 and clearly describing in the legend what the colors mean.  Most of the text in the legend (from Aspirin to bleeding risk) should be deleted. All this information is already in the main body of the review. As a minimum, a lot of unnecessary detail and verbosity should be removed.

Author Response

Title: The cellular and protein arms of coagulation in diabetes: established and potential targets for the reduction of thrombotic risk

We would like to take this opportunity to thank the expert reviewers whose time and suggestions have contributed to improving the manuscript and have strengthened the message. We have carefully revised the manuscript according to their comments and an itemised response to each comment and the changes as they appear in the manuscript are provided below.

Reviewer 2

We are grateful for the insightful comments of the reviewer. Their comments have been incorporated and considerably improved the manuscript.

Comment 1: This review addresses a timely topic, namely the prevention of cardiovascular complications of diabetes due to alterations of the haemostatic response. This review has three clearly delimited parts. The first part is a brief account of the physiology of the haemostasis process. It is a good introduction, more or less at textbook level, but not more is really needed to set the scene. This part is followed by a summary of the abnormalities in the process caused by diabetes. The last part is about current and prospective therapeutic options. The topic is timely, as I mentioned, and in continuous evolution, therefore this review is of value for researchers in the field. The review is well structured and generally well written, and the figures are informative and of good quality. There are nevertheless a number of issues that need to be addressed before I can recommend publication.

Response 1: Thank you for your kind words.

Comment 2: The last two paragraphs of the introduction feel disruptive. I suggest moving all this information to the appendix. It only needs a very short reference to the appendix at the end of the introduction.

Response 2: We agree with the reviewer and have therefore moved the final two paragraphs to “appendix 1” with a short reference in the final sentence of the introduction. Thank you.

Comment 3: Section 2, Mechanistic pathways of what? Please complete the title of this section.

Response 3: Thank you for identifying our error- we have amended this to “The mechanistic pathways of coagulation” with an additional sentence in 2.1 to highlight the link between coagulation and haemostasis.

Comment 4: Reference to figure 1 should appear much earlier, not nearly at the end of the section. In fact, each individual panel should be referenced in its corresponding subheading (Fig. 1A in section 2.1; Fig. 1B in section 2.2; Fig. 1C in section 2.3). Panel C could be more specifically dedicated to fibrin formation/fibrinolysis and, for completeness, section 2.3 should be expanded a bit. I miss things like urokinase and TAFI (this one is mentioned later, but it should have been introduced here), to mention a few.

Response 4: We agree with the reviewer’s insightful comment wholeheartedly. As such, we have referenced Figure 1A/B/C within the corresponding sections. We have also improved panel C to include the antifibrinolytic proteins, along with expansion of section 2.3 to discuss fibrin formation and fibrinolysis in more detail with the addition of urokinase and TAFI.

Comment 5: In 3.1, CD62P and P-selectin are the same thing but they are presented as activation marker first and as surface adhesion molecule next. Consider re-writing this part, but this is not critical.

Response 5: We thank the reviewer for their suggestion which has now been rephrased in the text to reflect the dual role of P-selectin.

Comment 6: In 3.2, 4th line, please complete: …expression of prostacyclin receptors (43) …

Response 6: Thank you, this has been amended in the text.

Comment 7: In the next paragraph, complete: …receptors result in fewer phosphorylated IRS-1 and…

Response 7: Thank you, this has been altered in the text.

Comment 8: In the same paragraph the authors mention nitrous oxide, they surely mean nitric oxide. The same happens on pg.5, near the end.

Response 8: We thank the reviewer for their diligence in reviewing the manuscript and catching this error. It has now been changed to “nitric” oxide which was our intention.

Comment 9: On pg. 5 paragraph 3, the LDL abbreviation was actually introduced earlier. Then you abbreviate VLDL but not HDL. In section 3.4 it may be advisable to include the acronym for SERCA.

Response 9: Again, we thank the reviewer for their sharp eyes. We have amended the abbreviations, choosing to remove “VLDL” altogether since it isn’t mentioned again in the manuscript. We agree with the reviewer regarding SERCA and this has been reflected in the text.

Comment 10: Section 4. Please complete the title of this section, Fibrin abnormalities in diabetes mellitus

Response 10: Thank you; this has been added to the title.

Comment 11: Figure 2 should be mentioned at the beginning of the relevant section rather than at the very end, so readers can look at it while reading through the information. Same with figure 3. I notice panel B of Figure 3 is not described in the legend. Panels A and B should be referenced separately in the text where they fit best.

Response 11: Thank you, we have referenced Figures 2 and 3 earlier in the respective paragraphs to improve the experience for readers. We have also added in the labels for Figure 3A and B as they appear in the text.

Comment 12: Last sentence of pg. 7, please complete: …the thromboxane synthesis pathway… the G-protein coupled receptor P2Y12 (clopidogrel…

Response 12: Thank you; this has been reflected in the text as suggested.

Comment 13: Table 1 needs careful consideration. First of all, it includes antithrombotic therapies but also anticoagulants, so either change the title or split into two (antithrombotics, anticoagulants). Clarify in the table heading what OD and BD means. For aspirin, complete: thromboxane A2 synthesis. Several of the therapies are described as No recommendation, so why are they included in a table of current therapies?. Perhaps clarify (as you indicate in the text) that some of them are used in acute coronary syndrome irrespective of diabetic state? I also notice that some (abciximab, eptifibatide, tirofiban, maybe a couple of them more) are not mentioned at all in the text.

Response 13: We thank the reviewer for their insightful comment; we have amended the table and separated into antiplatelet and anticoagulants. We have also clarified OD/BD and aspirin’s action. As rightly pointed out, we have removed the therapies with no recommendation from the table titled “current” therapies.

Comment 14: On pg. 9, end of 5.1, complete …TXA-2 synthesis.

Response 14: Thank you; this has been added to the text.

Comment 15: In 5.2 you use DAPT and CABG acronyms but do not introduce them a couple of sentences before.

Response 15: Thank you for spotting this; it has been amended in the text.

Comment 16: On page 10, thromboxane subheading, complete: …use of a P2Y12 inhibitor monotherapy… In the GPVI subheading, you mention collagen 3 “circulating” near the subendothelium; probably not the best word choice, I guess you mean exposed collagen, please re-write.

Response 16: Thank you for your insights; this has been rephrased.

Comment 17: In the Anticoagulant therapies section of pg 11 (mislabeled 2.1) for each component you include information about how they are altered in diabetes. In most cases this information is mentioned here for the first time. All this should have been included in section 4 and there would be no need to repeat. It may not be necessary to create subheadings for each component; the same applies for section 6.1

Response 17: Thank you for your insightful comment. We have included the information as suggested in section 4.

Comment 18: In general, when you mention things like described above/below it may be better to indicate exactly which heading or subheading you mean.

Response 18: The reviewer makes an excellent point; we have updated these instances in the text.

Comment 19: Figure 4, unless I missed it, is not referenced in the text. Some of the therapies shown here go with a different color but this is not explained in the legend. I suggest also coloring novel targets like C3, TAFI, PAI-1 and clearly describing in the legend what the colors mean.  Most of the text in the legend (from Aspirin to bleeding risk) should be deleted. All this information is already in the main body of the review. As a minimum, a lot of unnecessary detail and verbosity should be removed.

Response 19: We thank the reviewer for this helpful comment; we have added in the reference alongside a much-improved image that categorises targets into established and future therapies.

Round 2

Reviewer 2 Report

The authors have addressed all my comments satisfactorily, I have no objections for acceptance of this review in its present form.